

# Assessment of Precipitation Error Propagation in Multi-Model Global Water Resources Reanalysis

Md Abul Ehsan Bhuiyan[1], Efthymios. I. Nikolopoulos[1], Emmanouil. N. Anagnostou[1],

Clement Albergel[2], Emanuel Dutra[3], Gabriel Fink[4], Alberto Martinez de la Torre[5], Simon Munier[2], and Jan Polcher[6]

[1] Department of Civil and Environmental Engineering, University of Connecticut, Storrs, Connecticut, USA

[2] CNRM UMR 3589, Météo-France/CNRS, Toulouse, France

[3] Instituto Dom Luiz, Faculdade de Ciências, Universidade de Lisboa, Portugal

[4] Landesanstalt für Umwelt Baden-Württemberg (LUBW), Germany

[5] Centre for Ecology and Hydrology, Wallingford, UK

[6] Laboratoire de Météorologie Dynamique du CNRS/IPSL, Ecole Polytechnique, Paris, France

*Correspondence to*: Emmanouil N. Anagnostou (manos@uconn.edu)





# Abstract

This study focuses on the Iberian Peninsula and investigates the propagation of precipitation uncertainty,
and its interaction with hydrologic modelling, in global water resources reanalysis. Analysis is based on
ensemble hydrologic simulations for a period spanning 11 years (2000–2010). To simulate the
hydrological variables of surface runoff, subsurface runoff, and evapotranspiration, we used four land
surface models—JULES (Joint UK Land Environment Simulator), ORCHIDEE (Organizing Carbon and
Hydrology in Dynamic Ecosystems), SURFEX (Surface Externalisée), and HTESSEL (Hydrology-Tiled
ECMWF Scheme for Surface Exchange over Land)—and one global hydrological model, WaterGAP3
(Water–Global Assessment and Prognosis). Simulations were carried out for five precipitation products—
CMORPH, PERSIANN, 3B42 (V7), ECMWF reanalysis, and a machine learning-based blended product.
As reference, we used a ground-based observation-driven precipitation dataset, named SAFRAN,
available at 5 km/1 h resolution. We present relative performances of hydrologic variables for the different
multi-model/multi-forcing scenarios. Overall, results reveal the complexity of the interaction between
precipitation characteristics and different modelling schemes and show that uncertainties in the model
simulations are attributed to both uncertainty in precipitation forcing and the model structure. Surface
runoff is strongly sensitive to precipitation uncertainty and the degree of sensitivity depends significantly
on the runoff generation scheme of each model examined. Evapotranspiration fluxes are comparatively
less sensitive for this study region. Finally, our results suggest that there is no single model/forcing
combination that can outperform all others consistently for all variables examined and thus reinforce the
fact that there are significant benefits in exploring different model structures as part of the overall
modelling approaches used for water resources applications.





# 1. Introduction

Improved estimation of global precipitation is important to the analysis of continental water resources and dynamics. Over the past few decades, several studies have described the use of different precipitation algorithms to develop precipitation products (http://ipwg.isac.cnr.it/algorithms.html and
http://reanalyses.org) at high spatial and temporal resolution on a quasi-global scale and for different hydrological applications, such as flood early warning and control, and drought monitoring (Hong et al., 2010; Wu et al., 2012; and Vernimmen et al., 2011 amongst others). Precipitation estimates suffer, however, from various sources of error that consequently impact hydrologic investigations (Mei et al., 2015; Mei et al., 2016; Seyyedi et al., 2014, 2015; Bhuiyan et al., 2017, Nikolopoulos et al., 2013).

Over the last decade, an increasing number of studies have contributed to the development of global precipitation estimation (Pan et al., 2010; Beck et al., 2017a; Kirstetter et al., 2014; Carr et al., 2015; Dee et al., 2011) aiming at the overall improvement of the hydrological applications and global water resource reanalysis. Numerous models of varying complexity can be used to generate an array of hydrological
products from precipitation forcing datasets (Vivoni et al., 2007; Ogden et al., 1994; Carpenter et al., 2001; Borga, 2002; Schellekens et al., 2017). Different hydrological models have different applications depending on the spatial and temporal scales of interest, as well as the simulated variables of interest, such as subsurface runoff, surface runoff, and evapotranspiration. Past studies (Fekete et al., 2004; Biemans et al., 2009) have revealed that the uncertainty in simulated hydrological variables mainly
depends on the uncertainty in precipitation and model parametrisation, and suggested subsequent exploration of different model structures as part of the overall modelling approach.

So far there are several studies that have analysed uncertainty in precipitation forcing and its impact on hydrologic simulations by usually evaluating hydrologic simulations based on multiple forcing applied
on a single model (Falck et al., 2015; Bitew et al., 2012; Behrangi et al., 2011; Mei et al. 2016; Bhuiyan et al., 2018; Gelati et al., 2018 among others). On the other hand, there are also past studies that have evaluated the model structural uncertainty and its impact on hydrologic simulations, usually by analysing the simulation outputs from multiple models and a single forcing dataset (Breuer et al., 2009; Haddeland



et al. 2011; Gudmundsson et al., 2012; Smith et al. 2013; Huang et al. 2017; Beck et al., 2017b). However, fewer studies have been dedicated on the analysis of the integrated impact of both forcing and model uncertainty on hydrologic simulations and from the existing ones most of them were focused on a single hydrologic variable such as streamflow (see for example Qi et al. 2016), evapotranspiration (Vinukollu et al., 2011) or a given hydrologic index such as drought index (Prudhomme et al., 2014; Samaniego et al. 2017). Findings from these past investigations have demonstrated that both forcing and model structure uncertainty have a great impact on hydrologic predictions and therefore highlight that using multi-model/multi-forcing ensemble is a more appropriate path forward for advancing the use of hydrologic model outputs. This conclusion raises at the same time the need for better understanding, characterizing and quantifying the uncertainty associated to multi-model/multi-forcing hydrologic ensembles. Thus, a better understanding of the ensemble spread of precipitation and simulated hydrological variables is necessary to improve water resource management and planning. This additionally means that there is also a need to assess hydrologic uncertainty in more than a single variable to be able to have a better and more integrative view on the interaction between forcing uncertainty, model uncertainty and hydrologic variable of interest that will allow to make hydrologic predictions more effective for water resources applications at large.

This study builds upon a unique numerical experiment that was carried out, as part of the activities of the Earth2Observe project (Schellekens et al. 2017), to investigate the impact of precipitation uncertainty propagation and its dependence on model structure and hydrologic variable. In this investigation, we used different precipitation forcing datasets based on (i) reanalysis, (ii) satellite estimates, as well as (iii) a "combined" stochastic precipitation dataset (Bhuiyan et al., 2018). To consider model structure and parameters, we examined simulations from five state-of-the-art global-scale hydrological and Land Surface Models (LSMs). With regard to water cycle variables, we evaluated precipitation uncertainty propagation to surface runoff, subsurface runoff, and evapotranspiration fluxes. The study area for this investigation is the Iberian Peninsula, which has precipitation and climate variability due to complex orography influenced by both Atlantic and Mediterranean climates (Rodríguez-Puebla et al., 2001; de Luis et al., 2010; Herrera et al., 2010). The analysis comprised two main parts: (1) performance and

sensitivity evaluation of the different model/forcing scenarios and (2) precipitation uncertainty propagation to the hydrological variables. We analysed hydrological simulation with a comparative assessment of the hydrological products and provided a detailed analysis of uncertainty in hydrological simulations for the different global hydrological and land surface models used in the multi-model global
water resources reanalysis. Finally, we examined the performance of precipitation products in hydrological applications and potential uncertainty attributed to precipitation error propagations.

The paper is structured as follows: section 2 presents the different types of forcing datasets used for the study and section 3 details the methodology we used for our model development and hydrological model
analysis. Section 4 summarizes the hydrological results, section 5 discusses the results and section 6 draws conclusions from the research conducted.

## 2. Study area and forcing data

This study is focused on the Iberian Peninsula (Figure 1). The climate of the peninsula is primarily
Mediterranean, with mostly oceanic at northern and semiarid at southern parts. The topography in the Pyrenees varies from near sea-level to 3,500 meters. The meteorological forcing datasets used are described below.

### 2.1. Reference Precipitation (SAFRAN)

The reference precipitation dataset, hereafter referred to as SAFRAN (Système d'analyse fournissant des renseignements atmosphériques à la neige), was recently created by Quintana-Seguí et al. (2016, 2017) using the SAFRAN meteorological analysis system (Durand et al., 1993). Spatially, SAFRAN precipitation data are presented at hourly time scale on a regular grid of 5 km resolution, spanning 35 years and covering mainland Spain, Portugal and the Balearic Islands (Quintana-Seguí et al, 2016).
SAFRAN used an optimal interpolation algorithm (Gandin, 1966) to produce a quality controlled gridded dataset of precipitation which combines ground observations and outputs of a meteorological model

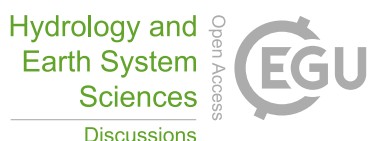

(Quintana-Seguí et al, 2017). On the other hand, several factors—including rainfall intermittency, discrete temporal sampling, and censoring of reference values for required quality—reduce the number of comparison samples for reference and satellite estimates. Therefore, the quality controlled SAFRAN dataset which is designed to force Land surface model is chosen as reference dataset for the study area
(Quintana-Seguí et al, 2017).

## 2.2. Satellite-Based Precipitation

Satellite-based simulations were based on three quasi-global satellite precipitation products. Among them CMORPH (Climate Prediction Center Morphing technique of the National Oceanic and Atmospheric Administration, or NOAA) is  developed from passive microwave (PMW) satellite precipitation fields
which is generated form motion vectors derived from Infrared (IR) data (Joyce et al., 2004). A neural network technique is used in PERSIANN (Precipitation Estimation from Remotely Sensed Information using Artificial Neural Networks) where IR observations are connected to PMW rainfall estimates (Sorooshian et al., 2000). Merged IR and PMW precipitation product from NASA are gauge adjusted for TMPA (Tropical Rainfall Measuring Mission Multisatellite Precipitation Analysis), or 3B42 (V7), which
is available in near-real time and post-real time (Huffman et al., 2010). The satellite precipitation products have spatial resolution is $0.25^0$ x $0.25^0$ and time resolution of 3-hourly.

## 2.3. Atmospheric Reanalysis

The reanalysis product (EI_GPCC) is based on original ERA-Interim 3-hourly data, after rescaling based
on GPCC (Global Precipitation Climatology Center) data. Note that total precipitation has been rescaled at monthly scale with a multiplicative factor to match GPCCv7 for the period 1979–2013 and GPCC monitoring for 2013–15. Data are further downscaled to $0.25^0$ x $0.25^0$ grid resolution by distributing the coarse grid precipitation according to CHPclim (Climate Hazards Group's Precipitation Climatology) high-resolution information for each calendar month. A similar approach was performed in the generation
of ERA-Interim/Land (Balsamo et al., 2015), but using GPCP (Global Precipitation Climatology Project). In this study we used GPCC due to its higher spatial resolution when compared with GPCP.



## 2.4. Combined Product

The combined product is based on the application of a nonparametric statistical technique for blending multiple satellite and reanalysis precipitation datasets. Specifically, a machine learning technique, Quantile Regression Forests (QRF) (Meinshausen, 2006), was used to generate stochastically an improved precipitation ensemble. The technique optimally merged global precipitation datasets and characterized the uncertainty of the combined product. Details on the methodology and data used to develop the combined product are presented in Bhuiyan et al. (2018).

## 2.5 Other atmospheric variables

Apart from precipitation forcing, the rest of atmospheric forcing variables required for the hydrologic simulations were derived from the original ERA-Interim 3hourly data as used in ERA-Interim/Land (Balsamo et al. 2015) bilinearly interpolated to 0.25°. It includes a topographic adjustment to temperature, humidity and pressure using a spatially-temporally varying environmental lapse rate (ELR) computed similarly to Gao et al (2012). The correction is the following: (i) relative humidity is computed from the uncorrected forcing; (ii) air temperature is corrected using the ELR and altitude differences (ERA-Interim topography versus 0.25 topography); (iii) surface pressure is corrected assuming the altitude difference and updated temperature; and (iv) specific humidity is computed using the new surface pressure and temperature assuming no changes in relative humidity.

## 3. Methodology

### 3.1 Hydrological Simulations

The hydrological simulations for this study were carried out by different collaborators within the framework of Earth2Observe, a European Union (EU) funded project using a number of global scale land surface/hydrological models. In this study, simulations from four land surface models—JULES (Joint UK Land Environment Simulator), ORCHIDEE (Organizing Carbon and Hydrology in Dynamic Ecosystems), SURFEX (Surface Externalisée), and HTESSEL (Hydrology Tiled ECMWF Scheme for Surface Exchanges over Land)—and one global hydrological model, the distributed global hydrological model of the WaterGAP3 (Water–Global Assessment and Prognosis) modeling framework (hereinafter





referred to as WaterGAP3) were considered. All models were forced with the various precipitation datasets described in the previous section for an 11-year period (March 2000–December 2010). A summary of some basic characteristics of the models structure is presented in Table 1 and a short description is provided below. For more details on the modelling systems, the interested reader is referred

to Schellekens et al. (2017) and references therein.

### 3.1.1 JULES

JULES (Best et al., 2011; Clark et al., 2011) is a physically-based land surface model. JULES uses an exponential rainfall intensity distribution to calculate throughfall through the canopy first (altered by interception), then the water reaching the surface is divided into infiltration into the soil and surface

runoff. Surface runoff is generated either through infiltration excess or saturation excess. Infiltration excess runoff will be generated by JULES if the water flux reaching the surface exceeds the maximum infiltration rate of the soil (based on the saturated hydraulic conductivity). Saturation excess runoff is based on subgrid soil moisture variability, as a fraction of the grid is saturated and water flux over this fraction is converted to surface runoff (Probability Distribution Model; Blyth, 2002). Once infiltrated into

the soil, water flows through the column resolved using Darcy's Law and the Richards' equation. Subsurface runoff is calculated using the free drainage approach, with water flowing at the bottom of the resolved soil column at a rate determined by the soil hydraulic conductivity. Further details on hydrology processes in JULES can be found in Best et al. (2011) and Blyth et al. (2018).

### 3.1.2 ORCHIDEE

ORCHIDEE (Krinner et al., 2005) is a complex land surface scheme that consists of a hydrological module, a routing (Ngo-Duc et al., 2007) and floodplain module (d'Orgeval et al., 2008). It also describes the vegetation dynamics and biological cycles but these features were not activated for the present study. The most relevant parametrisation of ORCHIDEE for the sensitivity of the model to rainfall is the one for partitioning between infiltration and surface runoff. In order to represent correctly the fast progression of

the moisture front during a rainfall event when the time step is above 15 minutes, a time-splitting procedure is used (d'Orgeval 2006). The parametrisation also takes into account reinfiltration in case of slopes below 0.5% or dense vegetation. Furthermore, as ORCHIDEE represents sub-grid soil moisture



by simulating separately the soil moisture column below bare soil, low and high vegetation, the infiltration process will display different sensitivities in each column.

### 3.1.3 SURFEX

The SURFEX modeling system of Météo-France (SURFace Externalisée, Masson et al., 2013) includes the ISBA LSM (Interactions between Soil, Biosphere, and Atmosphere, Noilhan and Mahfouf, 1996) that can be fully coupled to the CNRM (Centre National de Recherches Météorologiques) version of the Total Runoff Integrating Pathways (TRIP, Oki et al., 1998) continental hydrological system (Decharme et al., 2010). This study uses ISBA multi-layer soil diffusion scheme (ISBA-Dif) as well as its 12-layers explicit snow scheme (Boon et al., 2001, Decharme et al., 2016). ISBA total runoff is contributed by both the surface runoff and a free drainage as bottom boundary condition soil layer. The soil evaporation is proportional to its relative humidity. Furthermore, the Dunne runoff and lateral subsurface flow are computed using a topographic subgrid distribution.

### 3.1.4 WATERGAP31

The modelling framework WaterGAP3 is a tool to assess the global fresh water resources on 30-minutes spatial resolution. It combines a spatially distributed rainfall-runoff model with a large-scale water quality model as well as models for five sectorial water uses (Flörke et al., 2013; Döll et al., 2009). Effective precipitation – calculated as superposed effects of snow accumulation, snow melt and interception – is split into (i) a fraction that fills up an single-layer soil moisture storage and (ii) a fraction that comprises surface runoff and groundwater recharge. Groundwater recharge is the input of a single linear groundwater reservoir that is drained by base flow. Water for evapotranspiration, estimated with the Priestley-Taylor approach, is abstracted from the soil storage. The WaterGAP3 setting used in this study is calibrated and validated against measured river discharge from 2446 stations of the Global Runoff data Center data repository. Thereby, calibration only concerns the separation of effective precipitation into runoff and soil moisture filling. For a detailed model description see Eisner (2015).

### 3.1.5 HTESSEL

The land surface model (LSM) HTESSEL is part of the European Centre for Medium Range Weather Forecasts (ECMWF) numerical weather prediction model. The model represents the temporal evolution of the snowpack, soil moisture and temperature and vegetation water content, as well as the turbulent



exchanges of water and energy with the atmosphere. The soil column is discretized in four layers (7, 21, 72 and 189 cm thickness), and the unsaturated vertical movement of water follows Richards's equation and Darcy's law. The van Genuchten formulation is used to derive the diffusivity and hydraulic conductivity using 6 predefined soil textures. In case of partially or fully frozen soil, the water movement

in the soil column is limited by reducing the diffusivity and hydraulic conductivity. The model assumes free drainage as bottom boundary condition (sub-surface runoff) while the top boundary condition considers precipitation minus surface runoff and bare ground evaporation. Evapotranspiration is removed from the different soil layers following a prescribed root distribution (dependent on the vegetation type). Surface runoff generation is estimated as a function of the local orography variability, soil moisture state

and rainfall intensity. Soil saturation state and rainfall intensity define the maximum infiltration rate which is modulated by a variable infiltration rate related to orography variability (Balsamo et al 2009).

### 3.2. Evaluation metrics

To examine the magnitude and variability of differences among hydrological variables, we used relative
difference (RD) defined as:

$$RD = \left( \frac{\hat{y}_i - y_i}{y_i} \right), \qquad (1)$$

where $y_i$ denotes reference variables (SAFRAN-driven simulations) and $\hat{y}_i$ denotes simulated variables (based on the other forcing data considered) for each time step $i$. RD indicates the magnitude and direction of error with positive (negative) value indicating overestimation (underestimation).

To collectively assess the performance of various precipitation forcing datasets, models and simulated hydrological variables, we used a normalized version of the Taylor diagram (Taylor, 2001). Specifically, we normalized the values of the centered root mean square error (CRMSE) and the standard deviation



with the standard deviation of the reference. Therefore, the reference (that is, the target point to which the model outputs should be closest) corresponds to the point on the graph with the normalized CRMSE equal to zero, while both the correlation coefficient and normalized standard deviation equal one.

To evaluate the degree of variation of various precipitation datasets and simulated hydrological variables, we used coefficient of variation ($CV$) and coefficient of variation ratio ($CVr$). $CV$ is a measure of variability defined as the ratio of the standard deviation to the mean. To compare the degree of variation from one data series to another, we used $CV$ where we considered distributions with $CV < 1$ as low variance, while we considered those with $CV > 1$ as high variance. We defined $CVr$ as the ratio of the $CV$ 10  of model to the $CV$ of reference. The defined parameters are expressed as follows:

$$CVm = \frac{\sigma_m}{\bar{m}} \qquad (2)$$

$$CVo = \frac{\sigma_o}{\bar{o}} \qquad (3)$$

$$CVr = \frac{CVm}{CVo} \qquad (4)$$




$CVm$ and $CVo$ indicate coefficient of variation of model and coefficient of variation of reference, with the means $\bar{m}$ and $\bar{o}$ and standard deviations $\sigma_m$ and $\sigma_o$, respectively. The $CVr$ includes two components: the ratio of the means and ratio of the standard deviation. Details on the statistical metrics, including name conventions and mathematical formulas, are provided in the Appendix.

### 3.3. Metrics of uncertainty propagation

The random error component was based on the normalized centered root mean square error (NCRMSE). To demonstrate how error in precipitation forcing translates to error in the simulated hydrological variables—surface runoff (Qs), subsurface runoff (Qsb), and evapotranspiration (ET)—we used the NCRMSE error metric ratio as follows:

$$NCRMSE = \frac{\sqrt{\frac{1}{n}\sum_{i=1}^{n}\left[\hat{y}_i - y_i - \frac{1}{n}\sum_{i=1}^{n}(\hat{y}_i - y_i)\right]^2}}{\sqrt{\frac{1}{n}\sum_{i=1}^{n}(y_i - \bar{y})^2}}, \tag{5}$$

$$\alpha_{NCRMSE} = \frac{NCRMSE_{(variables)}}{NCRMSE_{(precipitation)}}, \tag{6}$$

where $NCRMSE$ is normalized centered root mean square error and $\alpha_{NCRMSE}$ is NCRMSE error metric ratio. The $\alpha_{NCRMSE}$ metric quantifies the changes in the random error from precipitation to simulated hydrological variables (Qs, Qsb, and ET) and can thus be used to assess magnification ($\alpha_{NCRMSE}$>1) or damping ($\alpha_{NCRMSE}$<1).



### 3.4. Analysis of Ensemble Spread

To assess how variability in precipitation ensemble translates to variability of the various hydrological simulations (Qs, Qsb, and ET) for the different modeling systems, we performed an analysis of ensemble spread (Δ) formulated as

$$\Delta = \frac{\sum(X_{max-}X_{min})}{\sum R}, \qquad (7)$$

in which $X_{max}$ and $X_{min}$ represent, respectively, the maximum and minimum of ensemble values at each time step, while $R$ is the corresponding value of the reference. Note that the members of ensemble constitute a sequence for each time step ($X_1$ , $X_2$ …….$X_{20}$ ). Therefore, the ensemble spread provides a measurement of the expected prediction intervals relative to the reference value and the larger the

ensemble spread (Δ) indicates relatively wider ranges of prediction intervals.

## 4. Results

### 4.1 Variability of Multiple Hydrological Model Simulations

To examine the magnitude and variability of the differences among both models and forcing datasets, we analysed the multi-model simulation results for three hydrological variables including surface runoff (Qs),

subsurface runoff (Qsb), and evapotranspiration (ET). Throughout this analysis, we used the SAFRAN-based simulation as the reference for comparison. Figures 2 to 4 present spatial maps of annual average values for each model, along with the relative differences of annual average estimates of the different hydrological variables for all the precipitation forcing datasets and models.

Examination of SAFRAN-based annual average values of surface runoff show that WaterGAP3, of which parametrisation is based (in present study) on a calibration that was done with WFDI forcing dataset (Weedon et al., 2014) and measured runoff data, estimates considerably higher surface runoff than the rest of the models particularly in the north and north-western part of the study area (Figure 2). Consequently, subsurface runoff (Figure 3) and evapotranspiration (Figure 4) from WaterGAP3 were





lower in that part of the study area. All these results display substantial differences in the spatial pattern of relative differences, which suggests that simulations are sensitive to both precipitation forcing and model uncertainty. Certain models seem to be more sensitive for given variables. For example, HTESSEL and ORCHIDEE are the models with the largest relative difference of Qs and both models exhibited

different behaviour, relative to the other models, when forced by the satellite precipitation. This suggests a distinct structural difference on the way precipitation is partitioned into surface/subsurface runoff between the two groups.

Looking at the variability of results for combined and reanalysis (EI_GPCC) forcing datasets, no

substantial differences occurred between reference and simulated surface runoff (Qs). However for the satellite-based simulations, there were significant deviations. Specifically, the CMORPH-based simulation showed significant overestimation for ORCHIDEE and HTESSEL, but this pattern was reversed for JULES, SURFEX, and WaterGAP3, an outcome that highlights the impact of model structure on precipitation error propagation.

For subsurface runoff, similar spatial patterns (with respect to Qs) were exhibited for the reference and the rest of simulations (Figure 3), which were also affected substantially by precipitation uncertainty. For example, looking at the different model simulations we can see that WaterGAP3 results reveal the lowest relative differences of Qs for almost all the precipitation forcings. In addition, CMORPH-based

simulation underestimated substantially for all the models. Figure 4 presents the spatial pattern of the results for evapotranspiration. For the combined product and EI_GPCC, results were consistent with low relative difference (<25%). On the other hand, CMORPH-based simulation showed an overall underestimation and deviated considerably from the results of the other precipitation products.

We also present a comparison of cumulative probability of the relative differences among precipitation

forcings (Figure 5) and the simulated hydrological variables (Figure 6). The distribution of relative differences, both in terms of type (denoted by the shape of the Cumulative Density Function-CDF) and magnitude, differed as a function of precipitation forcing, model, and the variable considered. The CDF of precipitation relative differences shows that CMORPH deviated significantly from the other



precipitation products (Figure 5). The surface runoff based on ORCHIDEE/HTESSEL displayed a clear separation of the CDF for combined product/EI_GPCC and satellite-based precipitation forcing (Figure 6). Specifically, it is interesting to note how 3B42 (V7) responds very differently from other precipitation forcing datasets for ORCHIDEE, highlighting again the sensitivity of runoff response to precipitation

structure (space/time variability) and its dependence on the rainfall-runoff generation mechanism.

Boxplots of the relative difference of different hydrological variables for the various forcing datasets/models at daily scale are shown in Figure 7. Note the inclusion of the relative difference of precipitation forcing to allow the comparison between relative differences in precipitation with that in the other hydrological variables. For each model, the boxplot shows a lower interquartile range (IQR),

marking lower variability for Qsb and ET compared to Qs. Results for combined product/EI_GPCC–based simulations showed less variability than the satellite based simulations. The SURFEX and WaterGAP3 exhibited the lowest variability compared to the other models. Overall, with the exception of few cases (e.g. 3B42(V7) for ORCHIDEE/HTESSEL and CMORPH for ORCHIDEE), uncertainty reduces progressively from precipitation to surface runoff, subsurface runoff and finally ET.

## 15  4.2 Performance of Multi-model Simulations

The normalized Taylor diagrams summarize the results for two different temporal scales. Figure 8 shows the results for the 3-hourly scale only for the two models with output available at that resolution (JULES and SURFEX), while Figure 9 presents results at daily scale for all five models. We aggregated the 3-hourly results from JULES and SURFEX to daily to compare them with the nominal daily output of

ORCHIDEE, WaterGAP3 and HTESSEL. Results improved with the temporal aggregation in reducing random error for JULES and SURFEX. As shown in the two figures, for the surface runoff the point representing 3B42 (V7) was always the farthest from the reference (NCRMSE>0.75), which means 3B42 (V7) was always associated with the worst performance. Simulations based on combined product/EI_GPCC were always consistent with significantly reduced NCRMSE values in the range of

0.25-0.8 for all the hydrological models. Results for simulated ET are more consistent among the various precipitation forcing datasets exhibiting normalised standard deviations in the range of 0.8-1.2. NCRMSE reduced significantly (<0.35) for each forcing dataset; accordingly, the correlation coefficient (CC) also



raised considerably (>0.9) showing a very high degree of agreement with reference-based simulations. For surface/subsurface runoff, SURFEX and WaterGAP3 models performed comparatively better than other models by reducing NCRMSE values especially for the combined product and EI_GPCC.

To illustrate the relative variability between precipitation and individual hydrological variables, we calculated the coefficient of variation ($CV$) and the coefficient of variation ratio ($CVr$) for all the hydrological models. To provide an understanding of the impact of precipitation uncertainty in hydrological simulations, we produced boxplots of $CV$ and $CVr$ for precipitation forcing datasets and individual hydrological variables for all the models, as shown in Figure 10. From the boxplots of $CV$ from reference-based simulations, the distributions of ET showed low variability ($CV < 1$), while the variability for all the other hydrological variables was high ($CV > 1$). In terms of $CVr$, the SURFEX model consistently displayed low medians, which were close to 1 for all the precipitation forcing datasets but CMORPH. Moreover, the variability of ET was much lower than that of the other variables examined and well captured in all the simulation scenarios.

A precipitation-forcing-wise comparison indicates that, the combined product/reanalysis underestimated precipitation variability more than other precipitation forcings, which affected the corresponding variability in Qs, for all the models except ORCHIDEE. Although there were no significant differences in terms of variability for combined product and reanalysis based simulations for the four models (JULES, SURFEX, WaterGAP3, and HTESSEL), substantial differences in variability between precipitation and Qs were observed for ORCHIDEE model. Satellite products overestimated precipitation variability, leading to overestimation of the variability of surface and subsurface runoff.

## 4.3 Assessment of Precipitation Error Propagation

To investigate the possible amplification, or dampening, of the precipitation error to the hydrologic variables examined, we quantified the NCRMSE error metric ratio ($\alpha_{NCRMSE}$) and results are demonstrated in Figures 11 and 12. For all the scenarios (at 3-hourly and daily scales) and almost all models, $\alpha_{NCRMSE}$ values were less than 1, which highlighted the damping effect on the random error of precipitation in simulated variables. In general, the damping effect increases (i.e. $\alpha_{NCRMSE}$ reduces) moving from surface to subsurface runoff and ET, highlighting once again the interaction between the



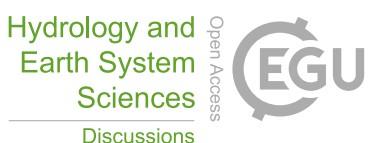
different runoff generating mechanisms as well as coupled water-energy balance processes and precipitation uncertainty. Interestingly, the relationship between error propagation among the different hydrologic variables varied greatly between models and precipitation forcing. Values of $\alpha_{NCRMSE}$ for surface and subsurface runoff are generally close for the SURFEX model but distinctly different for satellite-based results of ORCHIDEE and WaterGAP3.

## 4.4 Stochastic Precipitation Ensemble and Corresponding Simulated Hydrological Variables Analysis Results

The following summarizes the results of our analysis of ensemble precipitation (20 members), generated stochastically according to the algorithm used for the "combined" product, and their corresponding hydrological simulations. To show the relationship between the precipitation ensemble and simulated hydrological variables (generated ensemble), we presented an analysis of ensemble spread. Figure 13 depicts density plots between ensemble spread of precipitation and the simulated hydrological variables (Qs, Qsb, and ET) at monthly scale. A strong correlation between ensemble spread of Qs and precipitation is found for almost all models. For the other variables (ET, Qsb), ensemble spread was significantly narrower and rather independent of the ensemble spread of precipitation, manifested as the horizontal structure of contours in Figure 13. The ensemble spread of Qs was higher (ORCHIDEE and HTESSEL) or lower (SURFEX, WaterGAP3) depending on the model, elucidating again the impact of modelling structure on the propagation of precipitation uncertainty.

## 5. Discussion

Precipitation from different satellite and reanalysis datasets exhibits considerable differences in pattern and magnitude, which result in significant differences in hydrologic simulations. Results presented in this paper demonstrated clearly that magnitude and dynamics of uncertainty in hydrologic simulations depend not only on the uncertainty of the forcing variable, but also on the model and examined hydrologic variable.





For example, surface runoff (Qs) appears to be highly sensitive to precipitation differences, while ET was not for this semi-arid study region (Figures 2 to 4). Particularly, ET exhibited reduced sensitivity to precipitation forcing, which potentially suggests that the water volume available to be converted to ET did not deviate significantly among the precipitation scenarios. This is expected for ET, because, it is primarily controlled by atmospheric demand, plant and soil hydraulic constraints, and solar radiation (Wallace et al., 2010). When sufficient energy is available for rainfall to evaporate directly without contributing to surface/subsurface runoff, simulation of ET is not only affected by precipitation uncertainty, but also other atmospheric constrains.

Consequently, results (Figures 5 to 6) for ET were more consistent among the various model/precipitation forcing scenarios, indicating a smaller degree of uncertainty in ET (relative to Qs and Qsb). These results suggest that precipitation has a stronger influence on surface runoff, in particular precipitation intensity, i.e. the same amount of precipitation distributed over 3 hours or over 1 day will impact mostly surface runoff, and this is associated with the model representation of this fast process. Similarly, if we look at the distribution of precipitation relative difference, CMORPH tends to decrease in magnitude compared to other precipitation products. Therefore, for subsurface runoff, CMORPH-based simulations displayed a gross underestimation compared to other precipitation forcing.

Precipitation-to-surface runoff sensitivity is strongly controlled by the corresponding runoff generation scheme in each model. For example, in the case of HTESSEL and ORCHIDDE, precipitation intensity has a great effect on the generation of surface runoff. The satellite precipitation datasets have higher precipitation intensities (Figure 5), when compared to the remaining datasets, which explains the different behaviour of these two models. However, in the case of JULES, the infiltration excess mechanism is rarely invoked when the drivers are provided at a 3-hourly time step, as the maximum infiltration rate is not reached. Therefore, the significance of differences that HTESSEL and ORCHIDEE show with more intense rainfall are not shown by JULES due to distinct differences of their corresponding surface runoff generation modules.



Evaluation of the performance of the various simulations, relative to SAFRAN-based, emphasized the issues due to low correlation and increased random error from satellite products. On the other hand, the reanalysis (EI_GPCC) and combined product resulted in reduction of random error, suggesting that relying on gauge adjusted reanalysis or blended (satellite and reanalysis) products offers improvement

relative to satellite-alone products.

Certain dynamics resolved from this analysis were generally consistent among different models such as the fact that uncertainty reduced systematically from precipitation to surface runoff to subsurface runoff and eventually to ET simulations. This is also in accordance to our expectations given that soil moisture

(storage) integrates in time the precipitation variability. Surface runoff exhibits high correlation to precipitation while uncertainty in subsurface runoff is modulated by storage capacity of the soils. In addition ET is affected only if water availability deviates significantly from the water demand in terms of potential evapotranspiration. Our findings related to the surface runoff uncertainty (due to model structure and precipitation) suggest that the use of surface runoff (e.g. flash floods diagnostics) should be carefully

considered in each application in view of each model formulation.

## 6. Conclusions

This study investigated the propagation of precipitation uncertainty in hydrological simulations and its interaction with hydrologic modelling, which was based on satellite and reanalysis precipitation forcing

of a number of global hydrological and land surface models for the Iberian Peninsula. The following are the major conclusions from this study.

Simulation of surface runoff was shown to be highly sensitive to precipitation forcing, but the direction (that is, overestimation/underestimation) and the magnitude of relative differences indicated strong

dependence on the modeling system. Hydrological simulations based on reanalysis and combined product forcing datasets performed overall better than satellite precipitation–driven simulations. Moreover, simulation-results using CMORPH as forcing exhibit overall overestimation for ORCHIDEE/ HTESSEL





which is totally opposite to the results from the other models (JULES, SURFEX and WaterGAP3). These types of differences highlight the complexity of the interaction between precipitation characteristics and different modelling schemes and should be used as a "reference for caution" for when generalizing findings produced from single model simulations.

Modeling uncertainty appeared to be much less important for evapotranspiration than for surface and subsurface runoff. The sensitivity of hydrological simulations to different precipitation forcing datasets was shown to depend on the hydrological variable use and model parameterization scheme. Finally, based on our evaluation of the performance of the different hydrological models and five precipitation

products—CMORPH, PERSIANN, 3B42 (V7), reanalysis, and combined product—we could not identify a single model that consistently outperformed others i.e. certain models appeared more successful on the simulation of certain variables.

This study suggests important benefits may accrue from exploring different model structures as part of

the modeling approach. This study assessed the multi-model performances regarding three different hydrologic variables (surface/subsurface runoff, evapotranspiration). Apart from precipitation forcing, other atmospheric forcing variables required for the hydrologic simulations are also essential to investigate the significance of hydrological model uncertainty. In addition, the only calibrated model in this study WaterGAP3 performs better in specific locations (e.g., hilly) for all the hydrologic variables

than other models. Therefore, investigation should be performed in calibrating and regionalizing models for different parameters. Nevertheless, a clear outcome of the current work is that uncertainty in hydrologic predictions is significant and should be assessed and quantified in order to foster the effective use of the outputs of global land surface/hydrologic models. Considering ensemble representation (e.g. multi-model/multi-forcing) of hydrologic variables provides an appropriate path to address this issue.

Advancing our understanding on precipitation/model uncertainty and their interaction will potentially also aid in the investigation of the impacts of climate change (and associated uncertainty) on hydrological cycle components and water resource systems. Finally, this research provides a fine platform to discuss



advances in the applications of different precipitation algorithms, hydrology, and water resource reanalysis.

*Data availability.*

The datasets are available online for SAFRAN (https://doi.org/10.14768/MISTRALS-HYMEX.1388), CMORPH (ftp://ftp.cpc.ncep.noaa.gov/precip/CMORPH_V1.0/RAW/0.25deg-3HLY/), PERSIANN (http://fire.eng.uci.edu/PERSIANN/data/3hrly_adj_cact_tars/), 3B42(V7) (https://mirador.gsfc.nasa.gov), atmospheric reanalysis (https://wci.earth2observe.eu/portal/), satellite-derived near-surface daily soil moisture data (http://www.esa-soilmoisture-cci.org/node/145/), and the combined product (https://sites.google.com/uconn.edu/ehsanbhuiyan/research).

*Competing interests.* The authors declare that they have no conflict of interest.

*Acknowledgement.* This research was supported by the FP7 project eartH2Observe.

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



Figure 1. Map of Iberian Peninsula case study area.



Figure 2: Map of SAFRAN-based simulations (Reference) of surface runoff (top row) and relative difference for the various models (columns) and precipitation forcing (rows 2-5) analysed.





Figure 3: Same as Figure 2 but for subsurface runoff.





Figure 4: Same as Figure 2 but for evapotranspiration.

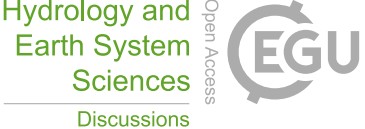



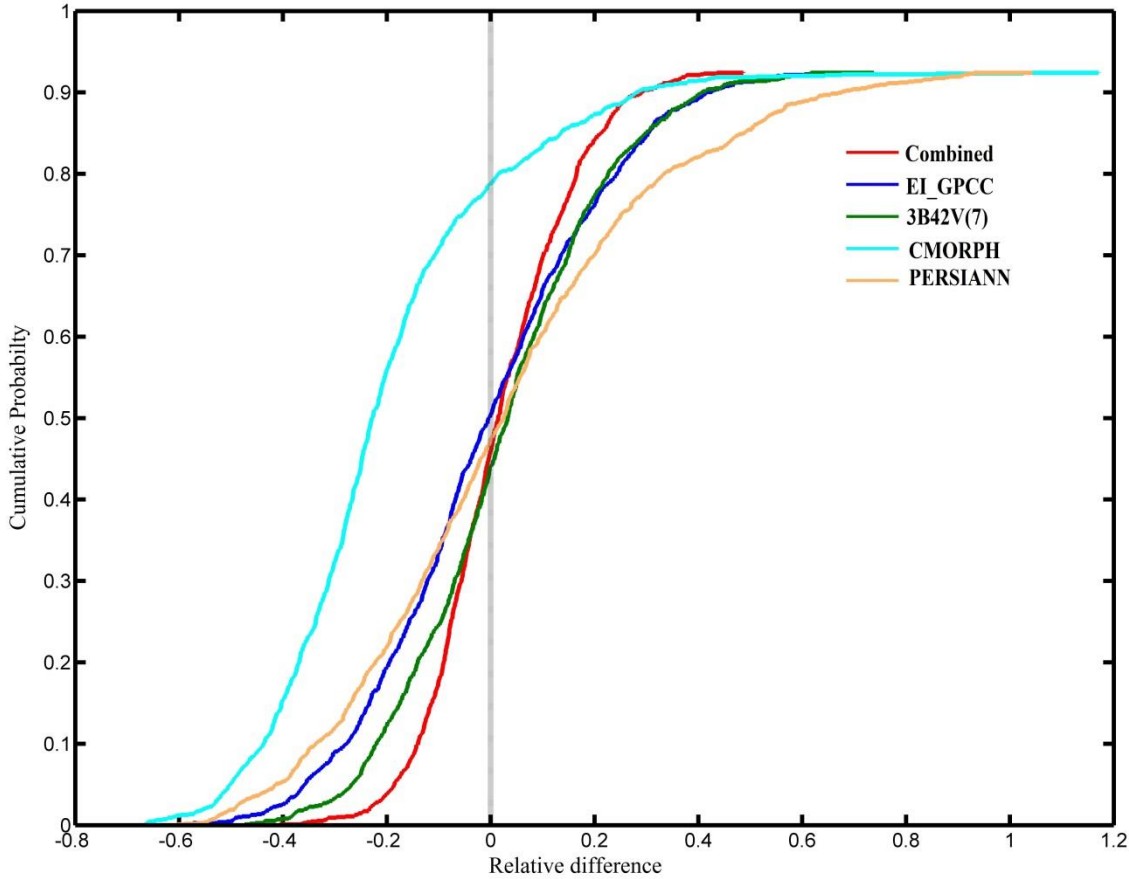

Figure 5: Cumulative probability for the precipitation forcing datasets.





Figure 6: Cumulative probability for the multi-model, multi-forcing simulations for simulated hydrological variables.





Figure 7: Relative difference presented for the various products and models at daily scale. In each box, the central mark is the median, and the edges are the first and third quartiles





Figure 8: Normalized Taylor diagrams for 3-hourly precipitation and simulated hydrological variables based on SAFRAN and the satellite/reanalysis precipitation products used.





Figure 9: Normalized Taylor diagrams for daily simulated hydrological variables with SAFRAN and the satellite/reanalysis precipitation products used.





Figure 10: Relationship between coefficient of variation and coefficient of variation ratio of simulated hydrological variables and precipitation.





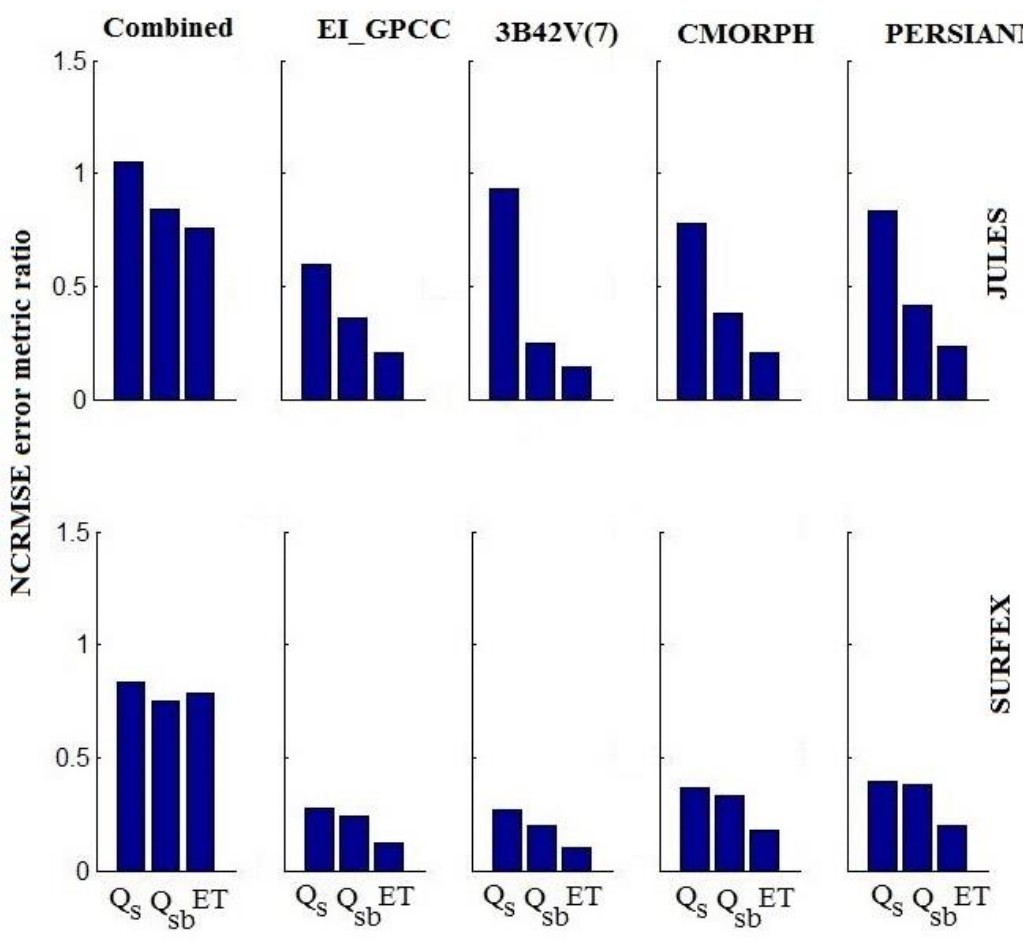

Figure 11: NCRMSE error metrics ratios presented for the various products and models at 3-hourly scale.





Figure 12: NCRMSE error metrics ratios presented for the various products and models at daily scale.





Figure 13: Density contour plot of the relationship between ensemble spread of simulated hydrological variables and precipitation at monthly scale. Color scale shows the frequency of occurrence. The black line is the 1:1 line.





**Appendix:**

The statistical metric, coefficient of variation ratio ($CVr$) used in the model evaluation analysis, was computed using following parameters:

$$\bar{o} = \frac{1}{N} \sum_{i=1}^{N} o_i \qquad (8)$$

$$\bar{m} = \frac{1}{N} \sum_{i=1}^{N} m_i \qquad (9)$$

$$\sigma_o = \sqrt{\frac{1}{N} \sum_{i=1}^{N} (o_i - \bar{o})^2} \qquad (10)$$

$$\sigma_m = \sqrt{\frac{1}{N} \sum_{i=1}^{N} (m_i - \bar{m})^2} \qquad (11)$$

Here, $o_i$ and $m_i (i = 1,...., N)$ are the observed and modeled time series of the product for times $i$, with the means $\bar{o}$ and $\bar{m}$ and standard deviations $\sigma_o$ and $\sigma_m$, respectively; and $N$ is the total number of

10   data points used in the calculations.



Table 1: Details on modeling systems.

| Model | Interception | Evapotranspiration | Soil layers | Ground water | Runoff | Reservoirs/ lakes | Routing |
|---|---|---|---|---|---|---|---|
| JULES | Single reservoir, potential evapotranspiration | Penman-Monteith | 4 | No | Saturation and infiltration excess | No | No |
| ORCHIDEE | Single reservoir structural resistance to evapotranspiration | Bulk ETP (Barella-Ortiz et al. 2013) | 11 | Yes | Green-Ampt infiltration | No | linear cascade of reservoirs (sub-grid) |
| SURFEX | Single reservoir, potential evapotranspiration | Penman-Monteith | 14 | Yes | Saturation and infiltration excess | No | TRIP with stream and deep-water reservoir at 0.5° |
| WATERGAP 3 | Single reservoir | Priestley-Taylor | 1 | Yes | Beta function | yes | Manning Strickler |
| HTESSEL | Single reservoir, potential evapotranspiration | Penman-Monteith | 4 | No | Saturation excess | No | CaMa-Flood |

