# Peer review of "Assessment of Precipitation Error Propagation in Multi-Model Global Water Resources Reanalysis"

_Hydrology and Earth System Sciences, 2018_

## Short Comment (SC1) · 2 Nov 2018

*Note to the editor and authors: As part of an introductory course to the Master pro-gramme Earth Environment at Wageningen University, students get the assignment to review a scientific paper. Since several years, students have been reviewing papers that are in open online discussion for HESS or BGS, and they have been asked to submit their reports to the discussion in order to help the review process. While these reports are written in the form of official (invited) reviews, they were not requested for by the editor, and we leave it up to the editor and authors to use these reports to their advantage. While several students were often asked to review the same paper, this*

[Figure]

*was not done with the aim to provide the authors with much extra work. We hope that these reports will positively contribute to the scientific discussion and to the quality of papers published in HESS. This report/review was supervised by dr. Ryan Teuling (teacher within the ITEE course at Wageningen University and also associated editor with HESS).*

The main objective of the article is the assessment of both uncertainty in precipitation forcing and in the structure of several land-surface models by simulating hydrological variables. Methodology proceeds by studying the relative differences of three simulated hydrological variables by running five state-of–the-art models each forced by six precipitation datasets of various source. I think there is a strong need of capturing the relative influence of uncertainty from as well the datasets as the model structure to increase efficiency in hydrological predictions for water resources and to explore the possible usefulness of multi-model/multi-forcing ensembles. Therefore, the assessment of integrated structural errors would be an addition to the scientific literature on this topic and would definitely be a topic of interest for HESS audience.

The paper is generally well-written in an understandable way, and the use of English is good. However, I my view the manuscript in its current state of is yet ready for publication and needs revision before it can be accepted. I hope the authors are willing to modify their manuscript, also taking into account the comments provided here. I feel the paper might benefit from a thorough restructuring to assure the manuscript meets the expected quality for HESS publications. Specific comments of three more major issues are described below, followed by discussion of a few minor issues

**Major recommendations**
The introduction starts very clear and leads correctly to a certain problem statement. Although, the research question and aim of the paper which are following, are very broad and besides I do not get a clear view of the specific objective of the authors. The integration between precipitation error and model structure error is important and should be assessed, but the question raised in this paper needs a more precise aim.

Momentarily, the aim of this paper leads to general results and a broad discussion and conclusion. For example, the conclusion partly states that the interaction between precipitation characteristics and different modelling schemes is very complex and the uncertainties in model simulations are due to precipitation and modelling structure errors. This conclusion is not a contribution to the scientific literature of this topic (Haddeland et al. 2011), while this subject has the competence to deliver a valuable product.

To be more precise, in the introduction a problem is sketched which needs attention but the following query is too broad. In my opinion, the paper should focus on a smaller and more specific subject to deliver an enhanced final product. The decision for coming to an alternative/specific aim is completely in hands of the original authors. Several suggestions to narrow this subject:

- Decreasing the number of precipitation forcings to one. As a consequence, more detailed results will be provided and a quantification of how the precipitation uncertainties behave or affect the different model-structures of the land-surface models. Via this way, the integration of the precipitation and model structure uncertainty will remain the main subject of the paper, but it will limit the scope.

- The other way around is also possible. By decreasing the number of land-surface models to one, the paper will show how the different precipitation forcings will influence the model structure uncertainty (in terms of hydrological variables).

- If, according to the authors, the essence of the article is damaged by reducing the number of forcings or models, the relative same procedure as before can be used. The disadvantage of this method is the huge amount of workload, because the results require a distinct end product, like quantifications of uncertainties, distinguished per model or per forcing.

The second concern is the main methodology of simulation. Although the selection of the several diverse precipitation products and the five state-of-the-art models is proper,

I do not understand why the one hydrological model is calibrated, but the four land-surface models are not. From my point of view, the outcomes of the simulations are only useful (in terms of considering model structure errors) when the models function in the same way, excluding the relative differences caused by alternative parameterization. This is also confirmed by Yin (2018), indicating that calibration of the models is necessary to use the same parameters while only the meteorological data is unique. To eliminate the effect of different parameterization, I expected the models to be calibrated to expose the uncertainties in model structures. Despite the fact that I am not considering myself an expert in the field of model calibration or validation, the main point here is the fact that this essential part of the methodology is rather unclear and additional information will be needed.

Thus, if the authors are able to verify that the current parameterization does not affect the results of the exposure of model structure error, the explanation of this issue is very important to include in the paper. If the results are possibly affected and the land-surface models do contain incomparable parameters, the assessment of model structure uncertainty may be inaccurate and therefore cannot be identified as a certified result. Then I would recommend to perform model calibration and consequently present the improved results. For inspiration of model calibration, I found very intriguing papers, such as the paper of Beven and Binley (1992) about the GLUE framework, and the paper of Clark (2008) about the modular framework FUSE, focussed on model structure.

The last issue is the presentation of the results. Although the approach of the results is very clear, the display of the graphs is quite cluttered and the amount is way too much. This issue relates to the first point of the review, providing the paper a more distinct aim. If the results would be presented more specific, the exact objective will be targeted much more efficient. At this moment, one does not know where to look for between 16 different graphs on one page and it gives the impression the reader needs to search for the results himself.

[Figure]

I would recommend to show only results that are specifically relevant to answer the research question, to prevent clustering of graphs and figures. Just use in the result section no more than one or two graphs of each sort of visual representation to support the corresponding findings. This recommendation does not even revolve around the possible change of the research question; even if the research question does not change, the authors should think of other ways of presenting their results.

**General comments**
One of the keywords of this paper is "model structure" and its respective uncertainty. However, the definition of this keyword is not thoroughly described and therefore different interpretations are allowed. In addition, there is a fine line between the concept of model structure and their corresponding parametrization. Thus, I would recommend to sharply define this keyword and so delineate its meaning for this paper.

I would recommend to add a line to the section concerning the study area why this area has been chosen. This is especially needed because in the discussion section (line 1-4) the authors insinuate that one of the hydrologic variables (evapotranspiration) is not the best measure of sensitivity for this study area. This is quite logical, because the study area is semi-arid and the amount of evapotranspiration is water-limited instead of energy-limited. Although the location of the study area is not damaging the research, in my opinion it would help to support the choice in this case.

**Further small remarks**
Page 4, line 20: Change "hydrologic variable" to "hydrologic variables".

Page 5, line 15: Change "semiarid" to "semi-arid" (for consistency, because at page 18, line 2, also "semi-arid" is used)

Page 6, line 4: remove the capital letter of 'Land surface model"

Page 7, line 11: Change "3hourly" to "3-hourly".

Page 8, line 15: Add comma after "column".

[Figure]

Page 12, line 13: Change "NCRMSE (variables)" to "NCRMSE (hydrologic variables)" or "NCRMSE (simulated variables)".

Page 17, line 23: Change "result" to "results".

Page 18, line 18: Change "ORCHIDDE" to "ORCHIDEE".

Page 32: Please add the full description for the graph, as on page 31.

Page 33: Please add the full description for the graph, as on page 31.

Page 37: Description of the Taylor diagram of SURFEX: change "maen sqaure" to "mean square" .

**References**

Beven, K. Binley, A. (1992). "The Future of Distributed Models: Model Calibration and uncertainty Prediction". Hydrological Processes, 6, 279-298.

Clark, M. (2008), "Framework for Understanding Structural Errors (FUSE): A modular framework to diagnose differences between hydrological models". Water Resources Research, 44(12).

Haddeland, I. (2011). "Multimodel Estimate of the Global Terrestrial Water Balance: Setup and First Results". Journal of Hydrometeorology, 12(5), 869-884.

Yin, Z. et al. (2018). "Comparing the Hydrological Responses of Conceptual and Process-Based Models with Varying Rain Gauge Density and Distribution". Sustainability, 10(9).
* * *

---

## Referee Comment (RC1) · Anonymous Referee #1 · 3 Nov 2018

Review of: "**Assessment of Precipitation Error Propagation in Multi-Model Global Water Resources Reanalysis**"

Authors: *Md Abul Ehsan Bhuiyan, Efthymios. I. Nikolopoulos, Emmanouil. N. Anagnostou, Clement Albergel, Emanuel Dutra, Gabriel Fink, Alberto Martinez de la Torre, Simon Munier, and Jan Polcher*

**General Comment:**

This paper compares outputs of four land surface models (LSMs) and a global hydrologic model (GHM) in the Iberian Peninsula forced by different precipitation (P) products for a period of 11 years. Precipitation products include satellite, reanalysis, and combined (stochastically generated) products. The SAFRAN precipitation products, which merge reanalysis and gauge observations, and the hydrologic simulations obtained with these precipitation inputs are assumed as reference.
The authors perform a set of analyses to evaluate how the uncertainties due to precipitation products and model structure affect three hydrologic variables, including surface runoff, subsurface runoff, ET.

The topics of the paper are interesting for the audience of HESS. The paper is, for the most part, well written. Thus, I am supportive of its publication. However, in my opinion, there are a few unclear parts in the text and analyses that require to be addressed first.

**Major concerns:**

1. It is not clear how the metrics used in the analyses are applied in terms of space and time aggregation. This should be clearly specified for each metric in the methodology section. For example, what is/are the time step/s of RD? How is this metric used in figure 5? Is figure 5 presenting the distribution of the RD's in all pixels (i.e., how is space considered)? Similar questions arise for the boxplots, the Taylor diagrams, and the CV. Please, clarify.

2. The time resolution of the satellite-based P products is 3 hours. How about the other two products? This, in combination with the resolution of the hydrologic simulations, affects the interpretation of the ability to simulate the hydrologic processes (notably, surface runoff).

3. The authors should provide in the Methodology section three details on the hydrologic simulations and their evaluation:
    (i) What is the time resolution adopted for each model?
    (ii) Was the model calibrated (I guess only one was) and, if not, which set of parameters was used?
    (iii) State that: (i) simulations are evaluated for long-term averages of annual, daily and, in some cases, 3-hour variables (see comment 1); and (ii) no seasonal analysis is performed.

4. The first result that I was expecting to see is the comparison of the bias between SAFRAN and the other P products (a figure like figure 2 but for P). This would give immediately an idea of what to expect for runoff and other hydrologic variables.

5. Related to the previous point: In my opinion, time series (at monthly resolution?) of spatially-averaged P, Qs, Qsb and E would be quite useful to have an idea of how the models vary among each other, across years, and within each year.

6. The analyses of the ensemble spread is not properly introduced in Section 3.4. What are the ensemble members referring to? Also, the definition of the metrics and associated symbols is not clear. Things become a bit clearer in section 4.4. However, I think that Sections 3.4, 4.4 and Figure 13 should be eliminated, since, as it stands, this analysis is superficial and does not add much to the message of the paper.

7. The interpretation on page 8, lines 21-23 is counterintuitive or I did not have enough information to understand it (see comment 3). To me, if an LSM is run at 3-hour resolution with a P product that has the resolution of 3 hours, there are higher chances that infiltration-excess runoff will be generated. This is because P products should be able to capture storms localized in time. In contrast, if an LSM is run at 3-hour resolution with a P product that has the original resolution of 24 hours and a uniform P intensity is assumed to create inputs at 3-hour resolutions, then the chances are lower.
On the other hand, if an LSM runs at 24-hour resolution and it has not been calibrated with P products at 3-hour resolution, then we can have unexpected effects on the generated runoff. In this case, I am not able to say a-priori what we should expect. Thus, the biases that the authors have found may be an effect of the calibrated parameters, rather than the model structure. I suggest the authors to clarify this part and elaborate more.

**Minor concerns:**

P 4, lines 12-16. This sentence seems too long.

P 5, lines 14-17: Please revise the sentences on climate and "topography in the Pyrenees". It doesn't make sense to me.

Section 2.1: Can the authors provide some quantitative information on the SAFRAN performances against rain gauges?

P6, line 4: consider using the acronym LSM for land surface model. Otherwise, don't capitalize "L".

P 15, line 22: I could not verify in the figure that NCRMSE > 0.75 for surface runoff simulated with 3B42 in all (or most) cases. Can the authors check this again and explain?

P. 16, lines 8-10: What are the implications of this? Is it expected? I could not figure this out by myself without knowing for which time scale the CV was computed.

P. 16, lines 10-13: I could not verify this interpretation in the figure. The median of the boxplots for SURFEX are for the most part larger than 1. Please, clarify.

Section 4.2: I suggest moving the sentence on page 16, lines 14-20 after line 8, as I believe that the comment on precipitation should be provided first.

Figure 7 should be improved. There are labels in the y axes only in some panels.

---

## Referee Comment (RC2) · Anonymous Referee #2 · 12 Nov 2018

The manuscript examines propagation of uncertainties in precipitation forcing (from satellite ‎and reanalysis) and in land surface models into simulation of hydrological variables, specifically, ‎surface runoff, subsurface runoff, and evapotranspiration. The study was conducted in the ‎Iberian Peninsula.‎ The importance of this study is in presenting the large uncertainties exist in both precipitation ‎and models, which induce substantial uncertainties in hydrological simulation. In accordance to ‎previous studies, it is shown in this work that precipitation uncertainties have the largest role ‎in prediction uncertainties, but the authors also show that there is a substantial uncertainties ‎originate from the model itself. This finding is important to be emphasized and to take into ‎account in hydrological simulations.‎

[Figure]

I have few suggestions for improvement:‎

‎1) I suggest adding a table comparing the different precipitation and reanalysis products, as ‎was done for the land surface models. Such a table should include information about the ‎resolution, and what data sets were used.‎

‎2) Sensitivity to product resolution: the different forcing products have different resolution, ‎which one could expect to affect the simulation results. It would be good to separate between ‎the uncertainty related to the product itself and the one related to its resolution, which may ‎too coarse for example for representing a given process. I suggest the authors to refer to this ‎aspect. ‎

‎3) There is almost no discussion of the role of the specific conditions in the Iberian Peninsula ‎and their relations with the findings. For example, it can be expected that surface runoff ‎sensitivity to precipitation uncertainties would be different in arid/semi-arid region compared ‎to mode humid areas. Since the study area includes a gradient of conditions, it would be good ‎to compare the different indexes among regions and possibly discuss this issue in Section 5.‎

‎4) What are the sources for additional data required for the models such as soil types, ‎groundwater table, others? ‎

Technical comments (typo errors and other):‎ P. 6 L. 4: "land" and "Land"‎ P. 6 L. 10: "from" instead of "form"‎ P. 7 L. 11: "3-hourly"‎ P. 8 L. 11-12: ". . . the water flux reaching the surface exceeds the maximum infiltration rate of ‎the soil". I believe the authors mean here the "final" infiltration, which is actually a minimum, ‎but not the maximal infiltration.‎ P. 9 L. 11: Please explain "Dunne runoff"‎ P. 10 L. 6: "a" is missing P. 13 Eq. 7: index i seems to be missing; why representing range by max – min and not std? ‎why not using "y" for reference?‎ P. 13 L. 10: please check, it is not clear

434, 2018.

---

## Author Comment (AC1) · 23 Dec 2018

**Response to Interactive discussion**

Hydrology and Earth System Sciences (HESS)

Title: Assessment of Precipitation Error Propagation in Multi-Model Global Water Resources Reanalysis

Md Abul Ehsan Bhuiyan1, Efthymios. I. Nikolopoulos1 , Emmanouil. N. Anagnostou1 , Clement Albergel2 , Emanuel Dutra3 , Gabriel Fink4 , Alberto Martinez de la Torre5 , Simon Munier2 , and Jan Polcher6

We would like to thank Reviewer for his insightful discussion and constructive suggestions. Below we provide a point-by-point response to his comments. Reviewer's comments are in red and our responses in black font.

General Comment:

This paper compares outputs of four land surface models (LSMs) and a global hydrologic model (GHM) in the Iberian Peninsula forced by different precipitation (P) products for a period of 11years. Precipitation products include satellite, reanalysis, and combined (stochastically generated) products. The SAFRAN precipitation products, which merge reanalysis and gauge observations, and the hydrologic simulations obtained with these precipitation inputs are assumed as reference. The authors perform a set of analyses to evaluate how the uncertainties due to precipitation products and model structure affect three hydrologic variables, including surface runoff, subsurface runoff, ET.

The topics of the paper are interesting for the audience of HESS. The paper is, for the most part, well written. Thus, I am supportive of its publication. However, in my opinion, there are a few unclear parts in the text and analyses that require to be addressed first.

Major concerns:

1. It is not clear how the metrics used in the analyses are applied in terms of space and time aggregation. This should be clearly specified for each metric in the methodology section. For example, what is/are the time step/s of RD? How is this metric used in figure 5? Is figure 5 presenting the distribution of the RD's in all pixels (i.e., how is space considered)? Similar questions arise for the boxplots, the Taylor diagrams, and the CV. Please, clarify.

In the revised paper we will include detailed information on the application of each metric and the associated spatio-temporal scales.

2. The time resolution of the satellite-based P products is 3 hours. How about the other two products? This, in combination with the resolution of the hydrologic simulations, affects the interpretation of the ability to simulate the hydrologic processes (notably, surface runoff).

The time resolution of all precipitation products (satellite, reanalysis and the combined products) is 3 hours. In the revised paper we will clearly specify the time resolution of each precipitation dataset, which should clarify the confusion.

3. The authors should provide in the Methodology section three details on the hydrologic simulations and their evaluation:

(i) What is the time resolution adopted for each model?

(ii) Was the model calibrated (I guess only one was) and, if not, which set of parameters was used?

(iii) State that: (i) simulations are evaluated for long-term averages of annual, daily and, in some cases, 3-hour variables (see comment 1); and (ii) no seasonal analysis is performed.

In the revised paper we will update section 3.1 Hydrological Simulations by providing the information requested by the reviewer.

4. The first result that I was expecting to see is the comparison of the bias between SAFRAN and the other P products (a figure like figure 2 but for P). This would give immediately an idea of what to expect for runoff and other hydrologic variables.

Thank you for the suggestion. In the revised paper we will add a new figure on the precipitation bias.

5. Related to the previous point: In my opinion, time series (at monthly resolution?) of spatially averaged P, Qs, Qsb and E would be quite useful to have an idea of how the models vary among each other, across years, and within each year.

Thank you for the suggestion. In the revised paper we will add a new figure which will provide those time series comparisons.

6. The analyses of the ensemble spread is not properly introduced in Section 3.4. What are the ensemble members referring to? Also, the definition of the metrics and associated symbols is not clear. Things become a bit clearer in section 4.4. However, I think that Sections 3.4, 4.4 and Figure 13 should be eliminated, since, as it stands, this analysis is superficial and does not add much to the message of the paper.

In the revised paper we will update detail about the analysis of the ensemble spread. Note that, the combined product is an ensemble based precipitation product; for the evaluations presented in this paper we use ensemble-mean as forcing. For the analysis and propagation of the precipitation ensemble spread to hydrologic simulations, we used 20 ensemble members, which are generated stochastically by the quantile regression forests (QRF) tree-based regression model (Meinshausen, 2006). We would like to keep these sections because they describe how variability in precipitation translates to variability of the various hydrological variables such as surface/sub surface and

evapotranspiration and how this varies across different models. This novel and complements well our multi-forcing/multi-model error analysis.

7. The interpretation on page 8, lines 21-23 is counterintuitive or I did not have enough information to understand it (see comment 3). To me, if an LSM is run at 3-hour resolution with a P product that has the resolution of 3 hours, there are higher chances that infiltration-excess runoff will be generated. This is because P products should be able to capture storms localized in time. In contrast, if an LSM is run at 3-hour resolution with a P product that has the original resolution of 24 hours and a uniform P intensity is assumed to create inputs at 3-hour resolutions, then the chances are lower.

On the other hand, if an LSM runs at 24-hour resolution and it has not been calibrated with P products at 3-hour resolution, then we can have unexpected effects on the generated runoff. In this case, I am not able to say a-priori what we should expect. Thus, the biases that the authors have found may be an effect of the calibrated parameters, rather than the model structure. I suggest the authors to clarify this part and elaborate more.

Thank you for this comment. In the revised paper we will clarify this issue.

Minor concerns:

P 4, lines 12-16. This sentence seems too long.

Thank you. It will be modified in the revised paper.

P 5, lines 14-17: Please revise the sentences on climate and "topography in the Pyrenees". It doesn't make sense to me.

Thank you. It will be updated in the revised paper.

Section 2.1: Can the authors provide some quantitative information on the SAFRAN performances against rain gauges?

Thank you. It will be updated in the revised paper.

P6, line 4: consider using the acronym LSM for land surface model. Otherwise, don't capitalize "L".

Thank you. It will be updated in the revised paper.

P 15, line 22: I could not verify in the figure that NCRMSE > 0.75 for surface runoff simulated with 3B42 in all (or most) cases. Can the authors check this again and explain?

In the revised paper we will update by proper information to clarify this issue.

P. 16, lines 8-10: What are the implications of this? Is it expected? I could not figure this out by myself without knowing for which time scale the CV was computed.

In the revised paper we will explain in detail to clarify this issue.

P. 16, lines 10-13: I could not verify this interpretation in the figure. The median of the boxplots for SURFEX are for the most part larger than 1. Please, clarify.

In the revised paper we will explain in detail to clarify this issue.

Section 4.2: I suggest moving the sentence on page 16, lines 14-20 after line 8, as I believe that the comment on precipitation should be provided first.

Thank you. It will be updated in the revised paper.

Figure 7 should be improved. There are labels in the y axes only in some panels.

Thank you. It will be corrected in the revised paper.

---

## Author Comment (AC2) · 23 Dec 2018

**Response to Interactive discussion**
Hydrology and Earth System Sciences (HESS)
Title: Assessment of Precipitation Error Propagation in Multi-Model Global Water Resources Reanalysis

Md Abul Ehsan Bhuiyan1, Efthymios. I. Nikolopoulos1 , Emmanouil. N. Anagnostou1 , Clement Albergel2 , Emanuel Dutra3 , Gabriel Fink4 , Alberto Martinez de la Torre5 , Simon Munier2 , and Jan Polcher6

We would like to thank Reviewer for his insightful discussion and constructive suggestions. Below we provide a point-by-point response to his comments. Reviewer's comments are in red and our responses in black font.

The manuscript examines propagation of uncertainties in precipitation forcing (from satellite and reanalysis) and in land surface models into simulation of hydrological variables, specifically, surface runoff, subsurface runoff, and evapotranspiration. The study was conducted in the Iberian Peninsula. The importance of this ˝ study is in presenting the large uncertainties exist in both precipitation and models, which induce substantial uncertainties in hydrological simulation. In accordance to previous studies, it is shown in this work that precipitation uncertainties have the largest role in prediction uncertainties, but the authors also show that there is a substantial uncertainties originate from the model itself. This finding is important to be emphasized and to take into account in hydrological simulations.

I have few suggestions for improvement:

1) I suggest adding a table comparing the different precipitation and reanalysis products, as was done for the land surface models. Such a table should include information about the resolution, and what data sets were used.

In the revised paper we will provide a new table describing the different precipitation products with necessary related information such as resolution, references etc.

2) Sensitivity to product resolution: the different forcing products have different resolution, which one could expect to affect the simulation results. It would be good to separate between the uncertainty related to the product itself and the one related to its resolution, which may too coarse for example for representing a given process. I suggest the authors to refer to this aspect.

In the revised paper we will clarify this issue. We would like to note that all precipitation forcing data were at 0.25 deg spatial resolution and 3-hourly temporal resolution.

3) There is almost no discussion of the role of the specific conditions in the Iberian Peninsula and their relations with the findings. For example, it can be expected that surface runoff

sensitivity to precipitation uncertainties would be different in arid/semi-arid region compared to mode humid areas. Since the study area includes a gradient of conditions, it would be good to compare the different indexes among regions and possibly discuss this issue in Section 5.

Very good comment. This aspect will be clarified in the revised manuscript.

4) What are the sources for additional data required for the models such as soil types, groundwater table, others?

In the revised paper we will include details about additional data required for the models.

Technical comments (typo errors and other):

P. 6 L. 4: "land" and "Land"

Thank you. It will be corrected in the revised manuscript.

P. 6 L. 10: "from" instead of "form"

Thank you. It will be corrected in the revised manuscript.

P. 7 L. 11: "3-hourly"

Thank you. It will be corrected in the revised manuscript.

P. 8 L. 11-12: ". . . the water flux reaching the surface exceeds the maximum infiltration rate of the soil". I believe the authors mean here the "final" infiltration, which is actually a minimum, but not the maximal infiltration.

Thank you. This sentence will be updated and modified in the revised manuscript to clarify this issue.

P. 9 L. 11: Please explain "Dunne runoff"

In the revised paper we will explain about Dunne runoff that would clarify this issue.

P. 10 L. 6: "a" is missing P. 13 Eq. 7: index i seems to be missing; why representing

range by max – min and not std? why not using "y" for reference?

Thank you. In the revised paper Eq. 7 will be updated based on reviewer's suggestion.

The maximum and minimum of ensemble values at each time step, indicate a comprehensive measurement (full coverage) of the expected prediction intervals relative to the reference value. Therefore, we chose ensemble range (Xmax – Xmin), instead of standard deviation.

P. 13 L. 10: please check, it is not clear

In the revised paper we will clarify.

---

## Author Comment (AC3) · 23 Dec 2018

Please see attached for our response to the reviewer.

Please also note the supplement to this comment:
https://www.hydrol-earth-syst-sci-discuss.net/hess-2018-434/hess-2018-434-AC3-supplement.pdf

---

## Author Response (AR1)

**Response to Interactive discussion: SC 1**

Hydrology and Earth System Sciences (HESS)

Title: Assessment of Precipitation Error Propagation in Multi-Model Global Water Resources Reanalysis

Md Abul Ehsan Bhuiyan, Efthymios. I. Nikolopoulos , Emmanouil. N. Anagnostou, Jan Polcher, Clement Albergel, Emanuel Dutra, Gabriel Fink, Alberto Martínez-de la Torre, and Simon Munier

We are glad that our work provided material for educational purposes and we hope that it triggered an
10  interesting discussion in the class. We would like to thank the Professor and his/her students for reading our manuscript and for providing insightful discussion and constructive suggestions. Below we provide a point-by-point response to these comments. Comments are in red and our responses in black font.

The main objective of the article is the assessment of both uncertainty in precipitation forcing and in the structure of several land-surface models by simulating hydrological variables. Methodology proceeds by
15  studying the relative differences of three simulated hydrological variables by running five state-of–the-art models each forced by six precipitation datasets of various source. I think there is a strong need of capturing the relative influence of uncertainty from as well the datasets as the model structure to increase efficiency in hydrological predictions for water resources and to explore the possible usefulness of multi-model/multi-forcing ensembles. Therefore, the assessment of integrated structural errors would be an
20  addition to the scientific literature on this topic and would definitely be a topic of interest for HESS audience. The paper is generally well-written in an understandable way, and the use of English is good. However, I my view the manuscript in its current state of is yet ready for publication and needs revision before it can be accepted. I hope the authors are willing to modify their manuscript, also taking into account the comments provided here. I feel the paper might benefit from a thorough restructuring to assure
25  the manuscript meets the expected quality for HESS publications. Specific comments of three

**Major recommendations**

The introduction starts very clear and leads correctly to a certain problem statement. Although, the research question and aim of the paper which are following, are very broad and besides I do not get a clear view of the specific objective of the authors. The integration between precipitation error and
30  model structure error is important and should be assessed, but the question raised in this paper needs a more precise aim. Momentarily, the aim of this paper leads to general results and a broad discussion and conclusion. For example, the conclusion partly states that the interaction between precipitation characteristics and different modelling schemes is very complex and the uncertainties in model simulations are due to precipitation and modelling structure errors. This conclusion is not a contribution
35  to the scientific literature of this topic (Haddeland et al. 2011), while this subject has the competence to

deliver a valuable product. To be more precise, in the introduction a problem is sketched which needs attention but the following query is too broad. In my opinion, the paper should focus on a smaller and more specific subject to deliver an enhanced final product. The decision for coming to an alternative/specific aim is completely in hands of the original authors. Several suggestions to narrow this subject:

• Decreasing the number of precipitation forcings to one. As a consequence, more detailed results will be provided and a quantification of how the precipitation uncertainties behave or affect the different model-structures of the land-surface models. Via this way, the integration of the precipitation and model structure uncertainty will remain the main subject of the paper, but it will limit the scope.

• The other way around is also possible. By decreasing the number of land-surface models to one, the paper will show how the different precipitation forcings will influence the model structure uncertainty (in terms of hydrological variables).

• If, according to the authors, the essence of the article is damaged by reducing the number of forcings or models, the relative same procedure as before can be used. The disadvantage of this method is the huge amount of workload, because the results require a distinct end product, like quantifications of uncertainties, distinguished per model or per forcing

The second concern is the main methodology of simulation. Although the selection of the several diverse precipitation products and the five state-of-the-art models is proper, I do not understand why the one hydrological model is calibrated, but the four landsurface models are not. From my point of view, the outcomes of the simulations are only useful (in terms of considering model structure errors) when the models function in the same way, excluding the relative differences caused by alternative parameterization. This is also confirmed by Yin (2018), indicating that calibration of the models is necessary to use the same parameters while only the meteorological data is unique. To eliminate the effect of different parameterization, I expected the models to be calibrated to expose the uncertainties in model structures. Despite the fact that I am not considering myself an expert in the field of model calibration or validation, the main point here is the fact that this essential part of the methodology is rather unclear and additional information will be needed. Thus, if the authors are able to verify that the current parameterization does not affect the results of the exposure of model structure error, the explanation of this issue is very important to include in the paper. If the results are possibly affected and the land-surface models do contain incomparable parameters, the assessment of model structure uncertainty may be inaccurate and therefore cannot be identified as a certified result. Then I would recommend to perform model calibration and consequently present the improved results. For inspiration of model calibration, I found very intriguing papers, such as the paper of Beven and Binley (1992) about the GLUE framework, and the paper of Clark (2008) about the modular framework FUSE,

focused on models tructure. The last issue is the presentation of the results. Although the approach of the results is very clear, the display of the graphs is quite cluttered and the amount is way too much. This issue relates to the first point of the review, providing the paper a more distinct aim. If the results would be presented more specific, the exact objective will be targeted much more efficient. At this moment, one does not know where to look for between 16 different graphs on one page and it gives the impression the reader needs to search for the results himself.

 I would recommend to show only results that are specifically relevant to answer the research question, to prevent clustering of graphs and figures. Just use in the result section no more than one or two graphs of each sort of visual representation to support the corresponding findings. This recommendation does not even revolve around

the possible change of the research question; even if the research question does not change, the authors should think of other ways of presenting their results.

References

Beven, K. Binley, A. (1992). "The Future of Distributed Models: Model Calibration and

uncertainty Prediction". Hydrological Processes, 6, 279-298.

Clark, M. (2008), "Framework for Understanding Structural Errors (FUSE): A modular

framework to diagnose differences between hydrological models". Water Resources

Research, 44(12).

Haddeland, I. (2011). "Multimodel Estimate of the Global Terrestrial Water Balance: Setup and First Results". Journal of Hydrometeorology, 12(5), 869-884.

Yin, Z. et al.  (2018). "Comparing the Hydrological Responses of Conceptual and Process-Based Models with Varying Rain Gauge Density and Distribution". Sustainability,10(9).

Thank you for your comments. Few studies have been dedicated on the analysis of the integrated impact of both forcing and model uncertainty on hydrologic simulations and from the existing ones most of them were focused on a single hydrologic variable such as streamflow/evapotranspiration. So, this paper uses the multi-forcing/multi-model experiment to address the following research questions:

1. How does the precipitation uncertainty propagate through the multi-model hydrologic simulations?

2. What is the relative importance of precipitation vs. modeling uncertainty on the simulation of key water cycle variables (surface/subsurface runoff and ET)?

3. What is the spread of the precipitation uncertainty in simulation of hydrological variables and how this depends on model type?

As mentioned above, this paper presents a unique precipitation-to-hydrologic simulations error analysis based on different hydrologic variables, multiple models and multiple precipitation datasets, to evaluate the role of uncertainty in precipitation forcing relative to modeling error. For this purpose, we considered multiple precipitation datasets and a number of global and land surface hydrologic models, which led to a comprehensive error propagation investigation. At the same time in our revised version we significantly expanded the model description, which will clarify the rationale of using both calibrated and uncalibrated hydrologic models. Moreover, the conceptional surface runoff generation process in WaterGAP3 was calibrated with measured river discharge because one of its initial purposes was to reproduce and assess current and past water resources – primarily in rivers. For the other models the focus was/is on the accurate reproduction of processes such as energy and mass fluxes at the surface, etc. For these physically well-defined processes, calibration is not necessary.

According to one of our research questions, how precipitation uncertainty propagates to the multi-model hydrologic simulations to assess hydrologic uncertainty in more than a single variable that will allow to make hydrologic predictions more accurate for water resources applications. These investigations provide quantification of the predictive uncertainty of multi-model/multi-forcing scenarios. Therefore, we would like to keep all these precipitation forcings and models to present relative performances of hydrologic variables for the different multi-model/multi-forcing.

**General comments**

One of the keywords of this paper is "model structure" and its respective uncertainty. However, the definition of this keyword is not thoroughly described and therefore different interpretations are allowed. In addition, there is a fine line between the concept of model structure and their corresponding parametrization. Thus, I would recommend to sharply define this keyword and so delineate its meaning for this paper. I would recommend to add a line to the section concerning the study area why this area has been chosen. This is especially needed because in the discussion section (line 1-4) the authors insinuate that one of the hydrologic variables (evapotranspiration) is not the best measure of sensitivity for this study area. This is quite logical, because the study area is semi-arid and the amount of evapotranspiration is water-limited instead of energy-limited. Although the location of the study area is not damaging the research, in my opinion it would help to support the choice in this case.

Thank you for your suggestion. In the revised manuscript, we substantially elaborated each model description which is related to model structure, additional data used, model parametrization etc. The reason behind choosing the study area is discussed in the introduction part: "The study area for this investigation is the Iberian Peninsula, which has precipitation and climate variability due to complex orography influenced by both Atlantic and Mediterranean climates (Rodríguez-Puebla et al., 2001; de Luis et al., 2010; Herrera et al., 2010)".

Further small remarks

Page 4, line 20: Change "hydrologic variable" to "hydrologic variables".

Thank you. It is corrected in the revised manuscript.

Page 5, line 15: Change "semiarid" to "semi-arid" (for consistency, because at page 18, line 2, also "semi-arid" is used)

Thank you. It is corrected in the revised manuscript.

Page 6, line 4: remove the capital letter of 'Land surface model"

Thank you. It is corrected in the revised manuscript.

Page 7, line 11: Change "3hourly" to "3-hourly".

Thank you. It is corrected in the revised manuscript.

Page 8, line 15: Add comma after "column".

Thank you. It is corrected in the revised manuscript.

Page 12, line 13: Change "NCRMSE (variables)" to "NCRMSE (hydrologic variables)" or "NCRMSE (simulated variables)".

Thank you. "NCRMSE (variables)" is replaced by 'NCRMSE (simulated variables)" in the revised version of the manuscript.

Page 17, line 23: Change "result" to "results".

Thank you. It is corrected in the revised manuscript.

Page 18, line 18: Change "ORCHIDDE" to "ORCHIDEE".

Thank you. It is corrected in the revised manuscript.

Page 32: Please add the full description for the graph, as on page 31.

Thank you. It is updated in the revised manuscript.

Page 33: Please add the full description for the graph, as on page 31.

5   Thank you. It is updated in the revised manuscript.

Page 37: Description of the Taylor diagram of SURFEX: change "maen sqaure" to "mean square"

Thank you. It is corrected in the revised manuscript.

**Response to Interactive discussion: Reviewer 2**

Hydrology and Earth System Sciences (HESS)

Title: Assessment of Precipitation Error Propagation in Multi-Model Global Water Resources Reanalysis

Md Abul Ehsan Bhuiyan, Efthymios. I. Nikolopoulos , Emmanouil. N. Anagnostou, Jan Polcher, Clement Albergel, Emanuel Dutra, Gabriel Fink, Alberto Martínez-de la Torre, and Simon Munier2

We would like to thank the Reviewer for his insightful discussion and constructive suggestions. Below we provide a point-by-point response to his comments. Reviewer's comments are in red and our responses in black font.

General Comment:

This paper compares outputs of four land surface models (LSMs) and a global hydrologic model (GHM) in the Iberian Peninsula forced by different precipitation (P) products for a period of 11years. Precipitation products include satellite, reanalysis, and combined (stochastically generated) products. The SAFRAN precipitation products, which merge reanalysis and gauge observations, and the hydrologic simulations obtained with these precipitation inputs are assumed as reference. The authors perform a set of analyses to evaluate how the uncertainties due to precipitation products and model structure affect three hydrologic variables, including surface runoff, subsurface runoff, ET.

The topics of the paper are interesting for the audience of HESS. The paper is, for the most part, well written. Thus, I am supportive of its publication. However, in my opinion, there are a few unclear parts in the text and analyses that require to be addressed first.

Major concerns:

1. It is not clear how the metrics used in the analyses are applied in terms of space and time aggregation. This should be clearly specified for each metric in the methodology section. For example, what is/are the time step/s of RD? How is this metric used in figure 5? Is figure 5 presenting the distribution of the RD's in all pixels (i.e., how is space considered)? Similar questions arise for the boxplots, the Taylor diagrams, and the CV. Please, clarify.

RD of annual average estimates of the precipitation forcing and different hydrological variables are calculated using daily datasets at the spatial resolution of 0.25◦. Moreover, cumulative probability of estimated annual average relative differences among precipitation forcings and the simulated hydrological variables are calculated using same spatial resolutions 0.25◦. Similarly, the normalized Taylor diagrams summarized model results for two different temporal scales (3-hourly and daily) at the spatial resolution

of $0.25^0$. CV and CVr are determined using all precipitation forcing and variables examined at $0.25^0$/daily resolution. In the revised manuscript, we included detailed information on the application of each metric and the associated spatio-temporal scales.

2. The time resolution of the satellite-based P products is 3 hours. How about the other two products? This, in combination with the resolution of the hydrologic simulations, affects the interpretation of the ability to simulate the hydrologic processes (notably, surface runoff).

The time resolution of all precipitation products (satellite, reanalysis and the combined products) is 3 hours. In the revised manuscript, we included table 1 which specifies the time resolution of each precipitation dataset, which will clarify the confusion.

3. The authors should provide in the Methodology section three details on the hydrologic simulations and their evaluation:

(i) What is the time resolution adopted for each model?

(ii) Was the model calibrated (I guess only one was) and, if not, which set of parameters was used?

(iii) State that: (i) simulations are evaluated for long-term averages of annual, daily and, in some cases, 3-hour variables (see comment 1); and (ii) no seasonal analysis is performed.

In the revised manuscript, we significantly reworked section 3.1, Hydrological Simulations, providing the information requested by the reviewer, which should clarify the confusion.

Specifically, the time resolution for each model is included in Table 2. Section 3.1 is updated by additional information with calibration information, parameters setting etc. The models were already evaluated at all time scales from daily to multi-annual.

4. The first result that I was expecting to see is the comparison of the bias between SAFRAN and the other P products (a figure like figure 2 but for P). This would give immediately an idea of what to expect for runoff and other hydrologic variables.

Thank you for the suggestion. In the revised paper, we added figure 2 to show the comparison of the bias between SAFRAN and the other P products.

5. Related to the previous point: In my opinion, time series (at monthly resolution?) of spatially averaged P, Qs, Qsb and E would be quite useful to have an idea of how the models vary among each other, across years, and within each year.

Thank you for the comment. We would like to show you the figures that provide those time series comparisons. In this response document, the monthly spatially averaged precipitation time series for different forcing are shown in Figure 1. Overall, individual satellite products overestimated precipitation, while the combined product and atmospheric reanalysis precipitation were more consistent relative to the reference precipitation. Similarly, the variability in the performance of different hydrologic simulated variables for different models and their inconsistencies relative to the reference are presented in Figure 2-4. Overall, there is no significant change in evapotranspiration within years. But, the monthly Qs shows overestimation for satellite precipitation forcing during the period of 2000-2010. Specifically, 3B42 (V7) based simulated Qs overestimated during the study period, which indicated poor performance compared to other forcing.

In our paper, to show the relative importance of precipitation and modeling uncertainty on the simulation of different variables, we focused our analysis mostly on two temporal scale (3-hourly and daily). Therefore, to maintain the consistency of the paper, we would like to exclude the monthly time series analysis results from the paper, although these results have great insight to get the idea of variability of precipitation forcing and hydrologic variables.

[Figure]

Figure 1: Time series of spatially averaged precipitation at monthly scale.

[Figure]

Figure 2: Time series of spatially averaged surface runoff at monthly scale.

[Figure]

Figure3: Time series of spatially averaged subsurface runoff at monthly scale.

[Figure]

Figure 4: Time series of spatially averaged evapotranspiration at monthly scale.

6. The analyses of the ensemble spread is not properly introduced in Section 3.4. What are the ensemble members referring to? Also, the definition of the metrics and associated symbols is not clear. Things become a bit clearer in section 4.4. However, I think that Sections 3.4, 4.4 and Figure 13 should be eliminated, since, as it stands, this analysis is superficial and does not add much to the message of the paper.

In the revised manuscript, we reworked the analysis of the ensemble spread. Note that, the combined product is an ensemble based precipitation product; for the evaluations presented in this paper we use ensemble-mean as forcing. For the analysis and propagation of the precipitation ensemble spread to hydrologic simulations, we used 20 ensemble members, which are generated stochastically by the quantile regression forests (QRF) tree-based regression model (Meinshausen, 2006). We would like to keep these sections because they describe how variability in precipitation ensembles translates to variability of the various simulated hydrological variables and thus provides quantification of the predictive uncertainty of the combined product-based simulations. Information on this predictive uncertainty and insight on its dependence on model structure and hydrologic variable is novel and informative for the potential use of approaches related to the combined product for probabilistic prediction of hydrologic variables.

7. The interpretation on page 8, lines 21-23 is counterintuitive or I did not have enough information to understand it (see comment 3). To me, if an LSM is run at 3-hour resolution with a P product that has the resolution of 3 hours, there are higher chances that infiltration-excess runoff will be generated. This is because P products should be able to capture storms localized in time. In contrast, if an LSM is run at 3-hour resolution with a P product that has the original resolution of 24 hours and a uniform P intensity is assumed to create inputs at 3-hour resolutions, then the chances are lower.

On the other hand, if an LSM runs at 24-hour resolution and it has not been calibrated with P products at 3-hour resolution, then we can have unexpected effects on the generated runoff. In this case, I am not able to say a-priori what we should expect. Thus, the biases that the authors have found may be an effect of the calibrated parameters, rather than the model structure. I suggest the authors to clarify this part and elaborate more.

Thank you for this comment. We indeed considered a time disaggregation of the rainfall from 3h to 15 minutes (as well as for other variables). But for rainfall it is much more difficult. We have chosen to spread the entire 3hourly rainfall over 1.5hours in these simulations because of sensitivity of modelled runoff. In the revised manuscript, the ORCHIDEE model description is elaborated with this aspect.

Minor concerns:

P 4, lines 12-16. This sentence seems too long.

Thank you. This sentence is modified in the revised paper.

P 5, lines 14-17: Please revise the sentences on climate and "topography in the Pyrenees". It doesn't make sense to me.

Thank you. It is updated in the revised paper.

Section 2.1: Can the authors provide some quantitative information on the SAFRAN performances against rain gauges?

Thank you. It is updated in the revised paper.

P6, line 4: consider using the acronym LSM for land surface model. Otherwise, don't capitalize "L".

Thank you. It is updated in the revised paper.

P 15, line 22: I could not verify in the figure that NCRMSE > 0.75 for surface runoff simulated with 3B42 in all (or most) cases. Can the authors check this again and explain?

In the revised paper, we updated by proper information to clarify this issue. As shown in Figure 10, the points for the 3B42 (V7) were always the furthest from the reference (NCRMSE>0.75) with low correlation coefficient (0.4-0.55) except SURFEX, which means 3B42 (V7) was always associated with the worst performance for all other models.

P. 16, lines 8-10: What are the implications of this? Is it expected? I could not figure this out by myself without knowing for which time scale the CV was computed.

From the boxplots of CV from reference-based simulations, the distributions of ET showed low variability (CV < 1), while the variability for all the other hydrological variables was high (CV > 1). Variability of ET is much lower than the other variables examined and it is well captured in all simulations scenarios This is expected for ET, because, it is primarily controlled by atmospheric demand, plant and soil hydraulic constraints, and solar radiation (Wallace et al., 2010). When sufficient energy is available for rainfall to evaporate directly without contributing to surface/subsurface runoff, simulation of ET is not only affected by precipitation uncertainty, but also other atmospheric constrains. CV was computed in daily scale.

P. 16, lines 10-13: I could not verify this interpretation in the figure. The median of the boxplots for SURFEX are for the most part larger than 1. Please, clarify.

In the revised paper, we explained precisely to clarify this issue. In terms of CVr, the SURFEX model performed very well by producing medians close to 1 (CVr=1, means equivalent degree of variability captured by the model) for all the precipitation forcing datasets but CMORPH.

Section 4.2: I suggest moving the sentence on page 16, lines 14-20 after line 8, as I believe that the comment on precipitation should be provided first.

Thank you. It is updated in the revised paper according to your suggestion.

Figure 7 should be improved. There are labels in the y axes only in some panels.

Thank you. It is corrected in the revised paper.

**Response to Interactive discussion: Reviewer 3**

Hydrology and Earth System Sciences (HESS)

Title: Assessment of Precipitation Error Propagation in Multi-Model Global Water Resources
Reanalysis

Md Abul Ehsan Bhuiyan, Efthymios. I. Nikolopoulos, Emmanouil. N. Anagnostou, Jan Polcher,
Clement Albergel, Emanuel Dutra, Gabriel Fink, Alberto Martínez-de la Torre, and Simon Munier

We would like to thank Reviewer for his insightful discussion and constructive suggestions. Below we provide a point-by-point response to his comments. Reviewer's comments are in red and our responses in black font.

The manuscript examines propagation of uncertainties in precipitation forcing (from satellite and reanalysis) and in land surface models into simulation of hydrological variables, specifically, surface runoff, subsurface runoff, and evapotranspiration. The study was conducted in the Iberian Peninsula. The importance of this ″ study is in presenting the large uncertainties exist in both precipitation and models, which induce substantial uncertainties in hydrological simulation. In accordance to previous studies, it is shown in this work that precipitation uncertainties have the largest role in prediction uncertainties, but the authors also show that there is a substantial uncertainties originate from the model itself. This finding is important to be emphasized and to take into account in hydrological simulations.

I have few suggestions for improvement:

1) I suggest adding a table comparing the different precipitation and reanalysis products, as was done for the land surface models. Such a table should include information about the resolution, and what data sets were used.

In the revised manuscript, we provided Table 1 describing the different precipitation products with necessary related information such as resolution, references etc.

2) Sensitivity to product resolution: the different forcing products have different resolution, which one could expect to affect the simulation results. It would be good to separate between the uncertainty related to the product itself and the one related to its resolution, which may too coarse for example for representing a given process. I suggest the authors to refer to this aspect.

We would like to note that all precipitation forcing data used were aggregated at $0.25^0$ spatial resolution and 3-hourly temporal resolution. Therefore, uncertainty related to precipitation forcing relates to the estimation uncertainty of each product and not with the space/time resolution.

3) There is almost no discussion of the role of the specific conditions in the Iberian Peninsula and their relations with the findings. For example, it can be expected that surface runoff sensitivity to precipitation uncertainties would be different in arid/semi-arid region compared to mode humid areas. Since the study area includes a gradient of conditions, it would be good to compare the different indexes among regions and possibly discuss this issue in Section 5.

We would like to thank the reviewer for this comment. We have revised the manuscript according to the following text to further discuss the spatial pattern of results.

"By examining the spatial pattern of relative differences (Figs 2-5) one can recognize that there is no consistent spatial pattern among the different model/forcing combinations. There are cases where the pattern of the differences is dominated by the pattern of precipitation differences, as for example the case of PERSIANN where the maximum of differences are concentrated in the central and eastern part of the peninsula. While there are other cases where the pattern is dominated by the sensitivity of the model (see for example results for ORCHIDEE/3B42 for surface runoff). "

4) What are the sources for additional data required for the models such as soil types, groundwater table, others?

In the revised paper, we included details about additional data required for the models in the model description section.

Technical comments (typo errors and other):

P. 6 L. 4: "land" and "Land"

Thank you. It is corrected in the revised manuscript.

P. 6 L. 10: "from" instead of "form"

Thank you. It is corrected in the revised manuscript.

P. 7 L. 11: "3-hourly"

Thank you. It is corrected in the revised manuscript.

P. 8 L. 11-12: ". . . the water flux reaching the surface exceeds the maximum infiltration rate of the soil". I believe the authors mean here the "final" infiltration, which is actually a minimum, but not the maximal infiltration.

Thank you. This section is significantly modified in the revised manuscript to clarify this issue.

P. 9 L. 11: Please explain "Dunne runoff"

In the revised paper, Dunne runoff is precisely defined that would clarify this issue.

P. 10 L. 6: "a" is missing P. 13 Eq. 7: index i seems to be missing; why representing

range by max – min and not std? why not using "y" for reference?

Thank you. In the revised manuscript, Eq. 7 is updated based on reviewer's suggestion.

The maximum and minimum of ensemble values at each time step, indicate a comprehensive measurement (full coverage) of the expected prediction intervals relative to the reference value. Therefore, we chose ensemble range (Xmax – Xmin), instead of standard deviation.

P. 13 L. 10: please check, it is not clear

In the revised manuscript the issue is completely modified.

[revised manuscript text omitted]